# Self-Healability of Poly(Ethylene-co-Methacrylic Acid): Effect of Ionic Content and Neutralization

**DOI:** 10.3390/polym14173575

**Published:** 2022-08-30

**Authors:** Nadim El Choufi, Samir Mustapha, Ali R. Tehrani-Bagha, Brian P. Grady

**Affiliations:** 1Chemical Engineering Department, American University of Beirut, Beirut 1107 2020, Lebanon; 2Laboratory of Smart Structures and Structural Integrity (SSSI), Department of Mechanical Engineering, American University of Beirut, Beirut 1107 2020, Lebanon; 3Department of Bioproducts and Biosystems, School of Chemical Engineering, Aalto University, 02150 Espoo, Finland; 4School of Chemical, Biological and, Materials Engineering, University of Oklahoma, Norman, OK 73104, USA

**Keywords:** self-healing polymers, mechanical properties, thermal properties, poly(ethylene-co-methacrylic acid), ionomer, projectile test, self-healing efficiency

## Abstract

Self-healing polymers such as poly(ethylene-co-methacrylic acid) ionomers (PEMAA) can heal themselves immediately after a projectile puncture which in turn lowers environmental pollution from replacement. In this study, the thermal-mechanical properties and self-healing response of a library of 15 PEMAA copolymers were studied to understand the effects of the ionic content (Li, Na, Zn, Mg) and neutralization percentage (13 to 78%) on the results. Differential scanning calorimetry (DSC), dynamic mechanical analysis (DMA), and tensile testing were used to study the thermo-mechanical properties of PEMAA copolymers while the self-healing response was studied using the projectile test. Puncture sites were observed using scanning electron microscopy (SEM) and the healing efficiency was quantitatively measured using the water leakage test. Five different self-healing responses were observed and correlated to ionic content and neutralization. At high neutralization, divalent neutralizing ions (Zn and Mg) that have stronger ionic interactions exhibited brittle responses during projectile testing. PEMAA samples neutralized with Mg and Li at low concentrations had a higher healing efficiency than PEMAA samples neutralized with Zn and Na at low neutralization. The PEMAA copolymers with higher tensile stress and two distinct peaks in the graph of loss factor versus temperature that indicate the presence of sufficient ionic aggregate clusters had improved healing efficiency. By increasing the neutralization percentage from 20% to 70%, the tensile strength and modulus of the samples increased and their self-healability generally increased. Among the investigated samples, the copolymer with ~50% neutralization by Li salt showed the highest healing efficiency (100%). Overall, the strength and elastic response required for successful self-healing responses in PEMAA copolymers are shown to be governed by the choice of ion and the amount of neutralization.

## 1. Introduction

Self-healing polymers can partially or fully recover their initial performance after being damaged without the need for significant human intervention and repair [1,2,3]. Methods of measuring the healing efficiency are specific for a given application yet the overall aim is to quantify how much of the original mechanical, thermal, and/or electrical performance is retained [4,5,6]. Through the ability to regain their performance, self-healing polymers play a crucial role in increasing service life, decreasing time and cost of maintenance as well as lowering environmental pollution from the use and replacement of polymers [7,8].

Self-healing abilities in polymers have already found use in several applications. Mechanical self-healing polymers are used in remote applications such as marine infrastructure, aircraft, and spacecraft [2,4]. Self-healing coatings have been manufactured at an industrial scale to inhibit rust and provide long-lasting insulation [9]. Newer classes of self-healing polymers that retain their thermal or electrical conductivity are utilized in biomedical applications, sensors, membranes, and soft robotics [10].

Self-healing polymers are classified into three main categories: capsular, vascular, and intrinsic [1,11,12]. *Capsular* self-healing polymer-based systems introduce healing agents in microcapsules that crack after getting damaged. The healing agent flows to the damaged site and eventually fills the crack in the polymer-based system [13,14,15,16]. Similarly, *vascular* self-healing polymer-based systems incorporate vessels and micro-channels that are refilled with healing agents instead of single-use microcapsules [17]. The major limitation of capsular self-healing polymer-based systems is that healing occurs once at the same zone as microcapsules cannot be refilled while the complexity of vascular self-healing polymer-based systems poses a limitation in scaling and structural integrity [18,19]. *Intrinsic* self-healing polymers are more sustainable as healing occurs due to bond reversibility, which allows for repeated healing cycles and does not require healing agents [20,21]. The introduction of self-healing polymers in various applications provides great potential for increasing their service life and decreasing their environmental impact.

Poly(ethylene-co-methacrylic acid) (PEMAA) copolymers neutralized with a metal salt is an ionomer; an ionomer contains repeat units of electrically neutral molecules and a small mole fraction, typically less than 10%, of molecules that can be neutralized with either an anion or cation [22]. Ionomers behave as elastomers at room temperature but can be melted and processed as thermoplastics at higher temperatures. The dual behavior is integral to its self-healing ability. The ionic pairs aggregate in tightly packed zones that contain mostly or only ionic material and form ionic aggregate clusters as shown in Figure 1 [23,24]. The ionic aggregate clusters form thermally reversible crosslinks while also increasing the strength and melt viscosity of the polymer [25,26]. Due to the unique ordering, a projectile passing through the material heats the impact region allowing the aggregate formation to self-heal the damaged site.

To understand the dependency of PEMAA self-healability on heat transfer, one of the earliest qualitative tests performed was conducting multiple damage types including sawing, cutting, puncturing, and projectile testing [27,28]. Cutting by a razor blade and puncturing by a nail yielded no observable healing. However, sawing and projectile testing which generated a sizeable amount of heat led to observable self-healing, which confirmed that the healing mechanism depends on heat. Projectile testing where a PEMAA film sample was shot with an air rifle produced the most consistent result of self-healing due to the high thermal generation from friction at impact.

One study examined the effect of neutralization on the self-healing response of PEMAA by conducting projectile testing on PEMAA copolymers neutralized at 30% and 60% with Na [28]. PEMAA neutralized at 30% Na exhibited more consistent and repeatable self-healing results at room temperature. At higher temperatures both PEMAA copolymers failed to exhibit successful self-healing as the PEMAA copolymers did not behave as elastomers to provide an elastic recovery. They proposed a two-stage healing mechanism: an instantaneously elastic response with thermal generation producing molten edges followed by a second stage of the molten edges sealing and solidifying the puncture site [27,28,29].

In another study, zinc stearate was added to PEMAA neutralized at 30% by Na. Cracks extending from the puncture site were observed and the addition of zinc stearate led to a decrease in both the elastic response in the first stage and the viscous flow in the second stage of the healing mechanism [30]. The affecting parameters to achieve successful healing during projectile testing were found to be the thickness of the PEMAA sample, bullet speed, and a ratio of sample thickness to bullet diameter [31]. For a bullet velocity of 180 m s^−1^, the cut-off ratio was 0.2 (mm of sample thickness/mm of bullet diameter) above which healing occurs and the study confirmed the importance of a successful elastic response for self-healing to occur.

Mechanical characterization tests such as tensile testing and DMA testing were used to understand the effect of neutralization on the morphological and mechanical properties that are attributed to the self-healing mechanism. DMA temperature sweeps confirmed the ability of PEMAA ionomers to behave as elastomers at low temperatures and as thermoplastics at high temperatures [30]. As expected, tensile tests conducted at higher temperatures than room temperature decreased Young’s modulus and increased the extension. A maximum modulus was reported for monovalent ionic content that was attributed to optimal packing of carboxyl groups and alkaline ions within ionic aggregates [32].

All previous studies on PEMAA’s self-healing ability used one PEMAA copolymer neutralized by 30% and 60% Na. While it was shown that low neutralization (30%) produced more successful healing responses, the studies were limited to a binary comparison between two copolymers and do not provide an optimum neutralization at which healing occurs. In addition, the effect of ion type and the base PEMAA copolymer on the self-healing mechanism has not been investigated in depth. This study analyzes the self-healing ability of PEMAA copolymers using 3 PEMAA base copolymers with four different neutralizing ions, Li, Na, Mg, and Zn, at neutralization percentages ranging from 13% to 78%. Thermal and mechanical properties shown to affect the self-healing response were characterized using DSC, DMA, and tensile testing. Projectile testing was conducted for all 15 PEMAA ionomers. The puncture sites are categorized based on the self-healing response and the healing efficiency was measured. Lastly, by linking key findings from the characterization tests and the projectile tests, new relationships between the self-healing mechanism and thermal and mechanical properties are developed.

## 2. Materials and Methods

### 2.1. Materials

A library of 15 PEMAA copolymers was supplied by the former DuPont Chemical Company (now part of Dow, USA) as shown in Table 1. Percentages represent the fraction of acid groups that have been neutralized by metal salts of Li, Na, Mg, or Zn. To compare the self-healing response with a traditional widely used polymer, low-density polyethylene (PE) was used with a melting point of 110 °C, melt index of 1.0 g/10 min, and a molecular weight of 50,000 g/mol.

The BMI measures the viscosity or the ease of flow of the unneutralized PEMAA copolymer by recording the weight of the base flowing through a die of a specific diameter in 10 min at a specific temperature and pressure. For all PEMAA copolymers, the testing method followed the standards of ASTM D-1238 [33]. As shown in Table 1, the different bases used had a BMI of 33, 122, or 190 g/10 min, and Table 2 lists the properties of the base acid copolymer.

### 2.2. Sample Preparation

Injection molding (Fox and Offord MK4-031, Peterlee, UK) was used for producing samples from PEMAA granules with various geometries: dog-bone samples for tensile testing, rectangular samples for DMA testing, and rectangular films for projectile testing as shown in Figure 2 with the corresponding dimensions. All samples were injected at 9 bar with a barrel temperature of 130 °C and a mold temperature of 80 °C. The molds (Figure 2a,b) were allowed to cool to 25 °C before the sample was removed and stored. The second mold was produced to obtain films of 1 mm thickness for projectile testing.

### 2.3. Methods

**Thermal Properties.** Samples of 3–6 mg were cut from prepared samples using a precision knife and accurately weighed before they were placed in hermetic aluminum pans. A DSC, TA Instruments model Q2000 (New Castle, DE, USA), was used to run thermograms consisting of heating from 0 °C to 150 °C, followed by a cooling cycle from 150 °C to 0 °C and then another heating cycle from 0 °C to 150 °C. A heating and cooling rate of 5 °C/min was used with a constant purge of 50 mL min^−1^ of nitrogen gas. No enthalpy calibration was used as the focus of the DSC thermograms was to obtain the melting temperature peaks.

**Mechanical Properties.** A mechanical testing machine (Instron model 5943, Norwood, MA, USA) with a load cell of 1 kN was used to obtain stress-strain diagrams and measure the tensile strength, the tensile strain, the Young’s modulus, the maximum elongation, and the maximum stress at break for the PEMAA copolymers. Dog-bone specimens were prepared per the specifications of ISO 527—1BA with a gauge length of 30 mm and thickness of 4 mm. Tensile tests were conducted with manual grips at a constant strain rate of 1 mm/s at room temperature.

The Perkin Elmer DMA model 8000 (Waltham, MA, USA) was used to obtain the DMA curves of the PEMAA copolymers. For DMA testing, the edges of the dog-bone specimen were cut resulting in a rectangular sample. Temperature sweeps were conducted from −50 °C up to 95 °C at a rate of 2 °C/min with a frequency of 1 Hz and a static force of 0.10 N. The samples were tested using a single cantilever fixture and a liquid nitrogen cooling system was used to reach −50 °C. DMA analysis provided the storage modulus and the loss factor, which measure the specimen’s ability to store energy elastically and the ratio of loss to storage modulus, respectively [34]. At least five specimens for each PEMAA copolymer underwent a DMA temperature sweep and the average storage moduli were extracted at 94.5 °C. Investigating the storage modulus at 94.5 °C is to understand the mechanical behavior near the melting point of the PEMAA copolymers which affects the viscous flow in the second stage of the healing mechanism.

**Aging Test.** The DSC and DMA analyses were used to understand how the thermal and mechanical properties of PEMAA copolymers changed with time after being injection molded. Rectangular samples from PEMAA-26%Li-122M were prepared on the same day via injection molding above the melting point of PEMAA copolymers. For thermograms using the same DSC parameters previously listed, specimens were cut from one rectangular sample and stored to be tested on day 1,3,7,14,21,31,42, and 62 from the date of injection molding. For temperature sweeps from 25 °C to 95° using the DMA, the rectangular samples were stored and tested on day 1,3,18,24,44 from the date of injection molding.

**Projectile Test.** PEMAA film samples were placed in a rectangular aluminum sample holder that firmly secured the specimen in place. The sample holder consisted of two aluminum plates with a length of 20 cm, a width of 11 cm, and a thickness of 1 cm each made using a computer numeric control machine. Both plates had a square hole with a side length of 1 cm cut out in the center. The sample holder held the PEMAA film in a stable position during the high impact that occurred in projectile testing. The square hole exposed a region of the sample that would be punctured during the projectile test. As shown in the schematic in Figure 3, using a pellet gun, a Diana P1000 EVO2 TH (Rastatt, Germany), the samples were shot from a distance of 1 m with a Tiger^®^ pellet (UK) of 4.5 mm diameter and a flat end.

To measure the speed of the pellet, a computer microphone was used to detect the moment of firing the pellet gun and the moment of impact. For sound analysis, code was developed on LabVIEW 2017 that took into consideration outside noise, the distance of the microphone from the sample, and the distance of the microphone from the gun. The average firing speed during the projectile test was found using the difference in time between the moment of firing and the impact and the distance from the pellet gun to the sample.

**Puncture Characterization.** Puncture sites were investigated using Tescan Lyra3 FIB-SEM (Brno, Czech Republic). The self-healing efficiency $\left(\eta_{eff}\right)$ was measured using the water leaking test adapted from [30] with slight modification. The setup consisted of the punctured film placed on top of a sealed graduated cylinder containing 5.5 mL of water with a diameter of 15 mm. The time needed for 5.5 mL of water to pass through the puncture site was recorded and the extent of healing was measured using Equation (1), where $t_{sample}$ is the time taken for the water to pass through the PEMAA sample under testing and $t_{PE}$ is the time taken for 5.5 mL water to pass through a puncture site for the PE sample without any observable healing. The measurements were repeated 5 times for each sample to obtain the average self-healing efficiency and its standard deviation.
(1)$$\eta_{eff}=\frac{t_{sample}-t_{PE}}{t_{sample}}\times 100$$

## 3. Results and Discussion

### 3.1. Effect of Aging on Thermal and Mechanical Properties

Figure 4 shows the DSC thermogram of PEMAA-53%Li-M122 after 30 days from preparation as an example. Melting occurred at ~94 °C in both heat cycles while an endothermic peak was observed at ~54 °C only during the first heating cycle. The lower temperature endothermic peak is the secondary crystal melting peak of polyethylene segments [35,36,37].

To understand the disappearance/appearance of the secondary crystal melting peak better, we conducted an aging test for one sample. Figure 5 shows the DSC thermograms of PEMAA-26%Li-122M samples at a different aging time from the date of injection molding. The secondary crystal melting peak around 50 °C shifted to the right by 5 °C from day 1 to day 62 and grew in amplitude with time. The peak never fully stabilized as seen in relation to the large amplitude and shift in the thermogram of an original pellet that was stored for ~10 years. However, there was relatively very little change in secondary crystallization after one month when comparing samples tested between 21 days, 31 days, and 42 days from the date of being injection molded. The primary crystal melting endotherm exhibited two peaks that gradually became one peak and surprisingly shifted to lower temperatures by 4 °C from day 1 to day 62; i.e., significant thinning is occurring in the primary crystals as secondary crystals form. The explanation for this counterintuitive behavior is that the total amount of chains in crystals increases to overcome the thermodynamic penalty associated with smaller crystal thicknesses (i.e., the polymer in secondary crystals increases partially at the expense of higher thicknesses). However, the multiple peaks exhibited in Figure 5 are not shown in the second heating cycle in Figure 4. Slow secondary crystallization explains the lack of multiple peaks in Figure 4 as the second heating cycle occurs only 1 h after reaching its melting point.

The mechanical properties of PEMAA copolymers were also tested by DMA analysis for 44 days. Figure 6 shows the storage modulus versus temperature of samples that underwent a temperature sweep using the DMA at different periods from the date of injection molding. The storage modulus was recorded from 25 °C to 95 °C for all runs. The storage modulus increased by more than three-fold from day 1 to day 44 and minor changes were observed between 24 days and 44 days after injection molding as shown in Figure 6. The storage modulus, which is the ratio of elastic stress to strain during dynamic mechanical analysis, measures the amount of energy that the specimen can store elastically [38]. The storage modulus of PEMAA copolymers increases and the specimen gets stiffer with time. The relatively minor changes in the secondary crystallization peak and the storage modulus occur in the same time frame of one month. The three-fold increase of the storage modulus with time showed that secondary polyethylene crystals play a significant role in shaping the storage modulus of PEMAA copolymers.

To minimize changes in thermal and mechanical properties due to changes in polyethylene crystallites during room temperature annealing, all PEMAA copolymers were prepared and stored for 30 days before thermal and mechanical characteristic tests and projectile testing. Such a waiting period of 24–31 days for stabilizing the thermal-mechanical properties of ionomers has also been suggested in other studies [27,28,30,31]. It is expected that the mechanical properties and self-healing response of the PEMAA samples change noticeably over time during the first month after the injection molding and stabilization.

### 3.2. Thermal Properties

The thermal characteristics of PEMAA copolymers were studied by DSC between 0 to 150 °C and the results are shown in Figure 7. The secondary crystal melting temperature (SCMT) for all PEMAA copolymers was between 51 °C and 58 °C, while the primary crystal melting temperature (PCMT) was between 94 °C and 98 °C. An increase in neutralization percentage resulted in a trend of increasing SCMT and decreasing PCMT. Higher neutralization of ionic content decreased PCMT because more ionic aggregate clusters in the copolymers lead to stronger and larger cross-linked regions [39] which in turn interfere more with polyethylene crystallization both kinetically and thermodynamically. SCMT increases with neutralization level because more material is available for secondary crystallization, i.e., less is found in primary crystals, which leads to higher thickness secondary crystals. SCMTs and PCMTs of PEMAA samples at 20% neutralization with a BMI of 122 were close to each other showing the minor effect of ion type (Li, Na, Zn, Mg). However, at higher neutralization percentages, clear differences in SCMTs or PCMTs for PEMAA copolymers with different ions show that the identity of the ions does play a role. Unfortunately, a sample neutralized with Li at the same 70–80% level was not available for comparison. Although data is limited, different BMIs do not look to give major differences in melting temperatures at the same neutralization percentages for the same ions.

### 3.3. DMA Results

Figure 8 shows that an increase in BMI led to a decrease in the storage modulus at 94.5 °C at a given neutralization percentage because the BMI is inversely proportional to the viscosity of the polymer [40]. At the same BMI, a higher neutralization percentage led to an increase in storage modulus at 94.5 °C. The storage modulus of PEMAA-53%Li at 94.5 °C was almost double that of any other PEMAA copolymer making it the stiffest polymer in the library. Interestingly, for a series of PEMAA copolymers neutralized by potassium, a maximum tensile modulus occurred between 40 and 50% neutralization [32].

Figure 9 shows the loss factor of PEMAA copolymers at various neutralization percentages. The copolymers with low neutralization percentage of around 20% have a single peak ($\beta^{\prime }\;$) which splits into two peaks (β and α) at higher neutralization percentages [41]. The single relaxation peak $\beta^{\prime }$ is attributed to the short-range motion of carboxylic and carboxylate groups and ionic pairs dispersed in the amorphous region [42]. At higher neutralization, β was attributed to relaxation in the amorphous non-ionic region, and α was attributed to regions associated with concentrated ionic aggregate clusters [41]. The split of $\beta^{\prime }$ into  β and α indicate separate relaxation temperatures of (1) the ion-depleted polymer domain, and (2) the ionic clusters with the adjacent polymer chain in their vicinity possessing restricted mobility [23,32]. PEMAA neutralized at ~20% by Li and Mg exhibited two separate peaks (Figure 9a,b), while PEMAA neutralized at ~20% by Na or Zn presented only one peak $\beta^{\prime }$ (Figure 9c,d), indicating that an insufficient concentration of ionic aggregate clusters had been formed. Only at higher neutralization percentages at ~75%, PEMAA neutralized by Na or Zn presented two distinct peaks indicating the presence of ionic aggregate clusters as shown in Figure 9c,d. In addition, as the neutralization percentage increased for all PEMAA copolymers under investigation, the temperatures between the two peaks β and α increased, and the peaks themselves became more distinct. The increase in the temperatures between the two peaks has been previously noted for PEMAA neutralized by Na [23,41] and we have confirmed it to occur for all remaining ionic content that was tested. PEMAA copolymers that exhibited the formation of two peaks in the loss factor also exhibited the highest storage modulus at 94.5 °C.

### 3.4. Mechanical Properties

Figure 10 shows a representative stress-strain curve of PEMAA copolymers neutralized by Mg while Table 3 summarizes tensile testing results obtained for all of the 15 PEMAA. By increasing the BMI, the viscosity and tensile strength of the PEMA copolymers are reduced as shown in Figure 10. Figure 10 also shows that an increase in neutralization percentage led to a decrease in the extension of PEMAA copolymers and an increase in Young’s modulus during tensile testing.

Table 3 shows that an increase in neutralization leads to an increase in Young’s modulus for Mg, Li, and Zn neutralized PEMAA and a decrease in Young’s modulus for Na neutralized PEMAA. PEMAA-71%Zn-M190 had the highest Young’s modulus of 200 MPa in comparison to PEMAA-78%Na-M122 and PEMAA-73%Mg-M122 even though the base copolymer had a higher BMI. Zn neutralized PEMAA copolymer showed the lowest Young’s modulus in comparison with other PEMAA copolymers with a BMI of 122 g/10 min neutralized at low percentages (~20). The modulus in these systems is controlled by the fractional crystallinity, the distribution of crystallite sizes, as well as the ionic aggregate characteristics which cause the seemingly contradictory trend of Zn-neutralized samples versus the other two valences. PEMAA-73%Mg-M122 has a much higher tensile stress and Young’s modulus by 37% and 87% in comparison to PEMAA-78%Na-M122 while at lower neutralization percentage the difference in values is not very significant. This is due to the energy of Mg ionic interactions being stronger than that of Na because Mg has a +2 valency compared to +1 of Na [43]. PEMAA-53%Li-M122 and PEMAA-57%Li-M190, the only copolymers with a neutralization within the range of ~50% showed Young’s modulus and tensile stress higher than that of PEMAA-73%Mg-M122. PEMAA-53%Li-M122 had the highest tensile stress at maximum load. This may be due to optimal packing within ionic clusters where higher percentages can disrupt the structure and lower the amount of polymer chains and secondary crystals with restricted mobility in the vicinity of ionic clusters [32]. Overall, as seen with Zn and Mg, the increased valency of the cation plays an important role in increasing the stiffness and strength of the PEMAA copolymers at high neutralization percentages; such results have been noted elsewhere [43,44]. Tensile testing also suggested an optimal neutralization percentage for Li-neutralized PEMAA copolymers.

### 3.5. Self-healing Response and Analysis

The main aim of the projectile tests was to observe which PEMAA copolymers could self-heal and to what extent. Projectile testing in a controlled setting as explained in Materials and Methods provides a good way to investigate the self-healing response of PEMAA copolymers when subjected to a high impact [28,31]. By comparing the results, a correlation between the self-healing response and the ionic content and neutralization percentage of PEMAA copolymers was developed. The high-impact responses of projectile testing are presented both through qualitative SEM images and quantitative healing efficiencies as described in the experimental section.

Previous studies have proposed that the self-healing mechanism during projectile testing occurs in a two-step process [27,28,30,31,39]. The first step is the elastic response, where the fired pellet stretches and punctures the polymer while simultaneously transferring heat to the local region where the projectile occurs. Due to the elastomeric behavior of ionomers, the stretched material responds with an immediate elastic response and snaps back to close the puncture site. The second step after the elastic response is the viscous response where the molten edges of the puncture site, due to the frictional heat generation from the bullet, reattach and close the puncture site. The second step is found in all thermoplastics where two samples will weld together given sufficient time and heat above the glass temperature [45]. What differentiates PEMAA from thermoplastics is that the second stage occurs directly after the first stage, showing that both elastomer and thermoplastic properties which ionomers possess are vital for successful self-healing.

Upon shooting the film samples of PEMAA copolymers, high-impact responses (HIR) were captured using SEM. Due to the different ionic content and neutralization percentages of the PEMAA copolymers tested, different responses were observed: (1) Circular holes (Figure 11) that are hollow impact sites where the polymer material was ejected and fully detached from the film upon impact; (2) Door-flap structures (Figure 12) that retained the polymer material upon impact yet did not fully seal resulting in a slightly open puncture site similar to an opened door-flap; (3) Brittle holes (Figure 13a,c) that presented non-uniform holes with cracks extending from the edges and their diameters were consistently smaller than that of circular holes; (4) Line fractures (Figure 13b,d, and Figure 14c) that are thin line fractures that occurred around the impact site; (5) Sealed HIR (Figure 14a,b) that presented fully closed impact sites where the self-healing mechanism was fully successful. Table 4 shows the PEMAA samples and the high-impact responses they exhibited.

The projectile tests conducted on PE, PEMAA-21%Na-M122, PEMAA-21%Zn-M122, and PEMAA-20%Zn-M33 consistently produced HIR of circular hollow holes with a diameter ranging from 3.3 to 3.6 mm. Figure 11b shows a representative image of the HIR hole produced during the projectile test of PEMAA-21%Zn-M122. In comparison to the HIR of PE shown in Figure 11a, the HIR diameter of PEMAA-21%Na-M122 showed no improvement while both 20%Zn-neutralized PEMAA copolymers showed an improvement by having a 13% retraction of the hole diameter. An elastic response occurred on the outer periphery of the puncture site yet the material at the puncture site was ejected. Based on the observations, PEMAA-21%Na-M122, PEMAA-21%Zn-M122, and PEMAA-20%Zn-M33 were not able to complete the first stage of elastic recovery in the self-healing mechanism.

For Li and Mg-neutralized PEMAA at low percentages and Na-neutralized PEMAA at high percentages, the HIR exhibited an elastic response with the material at the puncture site still attached yet not fully sealed. Figure 12 shows SEM images of PEMAA-20%Li-M122 as a representative of the door-flap response. Seven PEMAA copolymers were able to retain the material at the puncture site as observed in the SEM images in Figure 12 reflecting that the first stage of elastic recovery does occur in these PEMAA copolymers. However, as shown in Figure 12c, the puncture site is not fully closed, indicating that either the elastic recovery did not retract the material enough to be in contact and weld with the peripheral polymer domain or the viscous stage of the self-healing mechanism was not successful if not enough thermal energy was transmitted to heat the puncture site above the melting point.

For PEMAA neutralized by Zn and Mg at high percentages, two different HIR have exhibited: brittle holes and line fractures. The brittle holes of PEMAA-71%Zn-M190 and PEMAA-73%Mg-M122 are shown in Figure 13a,c respectively, while Figure 13b,d represent line fractures, respectively. The brittle holes had fractures extending from the edges of the hole, while the line fractures formed a fracture either around or directly extending from the center of the impact site. It is important to note that the inconsistent result of either a line fracture or brittle hole only occurred in Zn and Mg at high neutralization percentages. While the two responses seem to be very different, the cracks extending from the holes and the line fractures suggest that Zn and Mg-neutralized PEMAA copolymers at ~70% neutralization exhibit a brittle self-healing response. Na-neutralized PEMAA at a similar neutralization percentage (78%) presented a door flap response. The brittle response for Zn and Mg-neutralized PEMAA copolymers is attributed to their valency which at a high neutralization percentage present stronger ionic bonds leading to a restriction in mobility that can result in brittle holes or line fractures. The average diameter of the brittle holes was ~0.5–1.5 mm, which is significantly smaller than the diameter of the holes for the three ionomer samples that exhibited circular holes. The brittle response and the hollow hole suggest that both categories of HIR did not complete the first stage of elastic recovery in the self-healing mechanism yet PEMAA-71%Zn-M190 and PEMAA-73%Mg-M122 have less observable damage vs. the 20% neutralized materials. Kalista et al. stated that higher neutralization presented the least successful self-healing ability, yet this was only tested using PEMAA neutralized by Na at 30% and 60% [28]. Our results suggest that even if the first stage of self-healing does not occur, a higher neutralization percentage retains more polymer material at the impact zone.

Li-neutralized PEMAA copolymers at ~50% (PEMAA-58%Li-M190 and PEMAA-53%Li-M122) exhibited sealed sites as well as fracture lines. Figure 14a,b present sealed HIR on the exit side and entry side of the impact site, respectively. PEMAA-53%Li-M122 consistently produced sealed sites while PEMAA-58%Li-M190 also produced line fractures shown in Figure 14c. PEMAA neutralized by Li at ~50% was the only PEMAA copolymer to present healed specimens. To understand why PEMAA-58%Li-M190 and PEMAA-53%Li-M122 exhibited the most consistent self-healing response, the following section expands the discussion by examining the healing efficiency in relation to the ionic content and neutralization as well as their thermal and mechanical properties.

Healing efficiency and correlation with Thermo-mechanical Properties.

Figure 15 shows the healing efficiency as a function of neutralization percentage with the ionic content differentiated. The healing efficiency quantitatively confirmed the puncture sites observed microscopically and developed a scale to measure how ionic content and neutralization percentages influenced the self-healing ability. Figure 15 shows that the healing efficiency increases up to 100% as the neutralization percentage reaches 50% and then decreases with further neutralization. Previous studies which were limited to only two or three PEMAA copolymers presented a linear relationship between self-healing and neutralization percentages [28,39]. At low neutralization percentages, door-flap structures or hollow holes were found, while at high neutralization percentages brittle holes and line fractures were found. Neutralization at 50% provided a balance of strengthening the polymer without causing a brittle response and hence retaining its elastic response to retracting fully. To understand this balance, the rest of this section correlates the healing efficiency with the different self-healing responses and the thermal and mechanical properties found in the characterization tests.

Figure 15 shows that PEMAA-21%Na-M122, PEMAA-21%Zn-M122, and PEMAA-20%Zn-M33 which exhibited a circular hollow hole in projectile testing had a healing efficiency of 0%,13%, and 15%, respectively. The low healing efficiencies and the circular holes after impact (Figure 11) produced by PEMAA neutralized by Na and Zn at low percentages could be attributed to the lack of sufficient ionic aggregate clusters as shown by the lack of two peaks in the loss factor obtained from the DMA results in Figure 9c,d. In other words, the three PEMAA copolymers do not possess the unique ability of ionomers to provide the immediate elastic response during the first step in self-healing which occurs due to the presence of sufficient ionic aggregate clusters that form crosslinks in the polymer domain and restrict mobility [39]. Such a finding showed that there was a minimum threshold of neutralization needed for the formation of enough ionic aggregate clusters to influence projectile impact resistance and that the loss factor in DMA experiments can be used to test for this threshold. These three copolymers also exhibited the lowest maximum load and tensile stress at maximum load during tensile testing by a minimum of 26% and 10%, respectively, in comparison to the remaining copolymers. The lack of ionic aggregate clusters in the three copolymers also affects the mechanical properties since they could not support a similar high load while being stretched without failing in comparison to other copolymers during the tensile tests.

However, Li and Mg cations show better performance at low neutralization (20%) as they outperformed Na and Zn. The presence of ionic aggregate clusters which can be observed in the loss factor versus temperature (Figure 9) using DMA temperature sweeps can be used to predict a high healing efficiency at low neutralization percentages.

The seven PEMAA copolymers (PEMAA-13%Li-M33, PEMAA-20%Li-M33, PEMAA-26%Li-M122, PEMAA-20%Na-M33, PEMAA-78%Na-M122, PEMAA-21%Mg-M33, and PEMAA-20%Mg-M122) that exhibited the door-flap structure (Figure 12) during projectile testing resulted in healing efficiencies ranging from 66% to 85%. Even though only the elastic recovery stage of the two-step self-healing mechanism occurred, the healing efficiency showed substantial improvement as these PEMAA copolymers were able to retain the material attached at the impact site. To retain the material, the PEMAA copolymers need to have enough strength at impact. To find a correlation between mechanical properties obtained in characterization tests and the self-healing response, Figure 16 and Figure 17 show the healing efficiency in relation to tensile stress at maximum load and the storage modulus at 94.5 °C, respectively. Figure 16 shows that an increase in tensile stress at maximum load increases the healing efficiency. All PEMAA copolymers which retained the material at the impact site had tensile stress at a maximum load higher than 15 MPa.

The success of self-healing for PEMAA-58%Li-M190 and PEMAA-53%Li-M122 can be attributed to having the highest tensile stress at maximum load (as shown in Figure 16) and high zero-shear viscosity [44]. Najm et al. reported that PEMAA-40%Li had the highest zero shear viscosity of 119 kPa × s in relation to PEMAA-69%Na and PEMAA-60%Zn which had zero shear viscosities of 86 and 58 kPa × s, respectively. This combination allows the PEMAA copolymers to perform the immediate elastic recovery with enough thermal energy generated at the impact to seal the puncture site. Previous studies showed successful self-healing at 30% while in this study PEMA neutralized ~50% outperformed both PEMAA neutralized at 20% and 70%. PEMAA neutralized ~50% displays optimal viscous and strength improvements yet not neutralized high enough to present brittle results or low enough to only perform partial elastic recovery seen in the door flap structure. Such a finding demonstrates that the self-healing mechanism in PEMAA is a balance of elasticity, viscosity, and strength impacted by the choice of ion and amount of neutralization. Figure 16 and Figure 17 show that an increase in tensile stress and storage modulus is correlated with an increase in healing efficiency. However, at mid-range and high neutralization (~50% and ~75%), characteristic thermal and mechanical tests were not able to accurately predict whether the self-healing will be the brittle response of a successfully sealed response.

## 4. Concluding Remarks

Thermo-mechanical and self-healing properties of a series of ionomers (PEMAA) with various ionic content and neutralization percentage were studied. The results of DSC and DMA showed that the thermal and mechanical properties of PEMAA change relatively slowly after 30 days from the injection molding. The formation of secondary crystals over time increased the storage modulus of PEMAA copolymers as expected.

At 20% neutralization, PEMAA neutralized by Li and Mg had significantly higher healing efficiencies than Na and Zn-neutralized PEMAA due to the presence of more ionic aggregate clusters as evidenced by a separate loss factor peak in DMA temperature sweeps.

At high neutralization percentages of 70% and above, DMA and tensile testing showed that the PEMAA polymers become increasingly stiff and brittle. The PEMAA copolymers at higher neutralization had higher tensile strength, storage modulus, and Young’s modulus while they also exhibited lower elongation at break. In projectile testing, the thermo-mechanical properties of PEMAA neutralized at high percentages produced puncture structures of line fractures and holes of 0.5–1.5 mm during projectile tests. All PEMAA neutralized at high percentages ~70% outperformed PEMAA neutralized at low percentages ~20% by a 10 to 50% increase in the healing efficiency.

Mid-range neutralized PEMAA samples were able to self-heal and seal the puncture site after being shot with an air rifle. This behavior was attributed to their viscoelastic properties to allow enough elasticity to allow an instantaneous snapback mechanism where the molten edges sealed the puncture site.

## Figures and Tables

**Figure 1 polymers-14-03575-f001:**
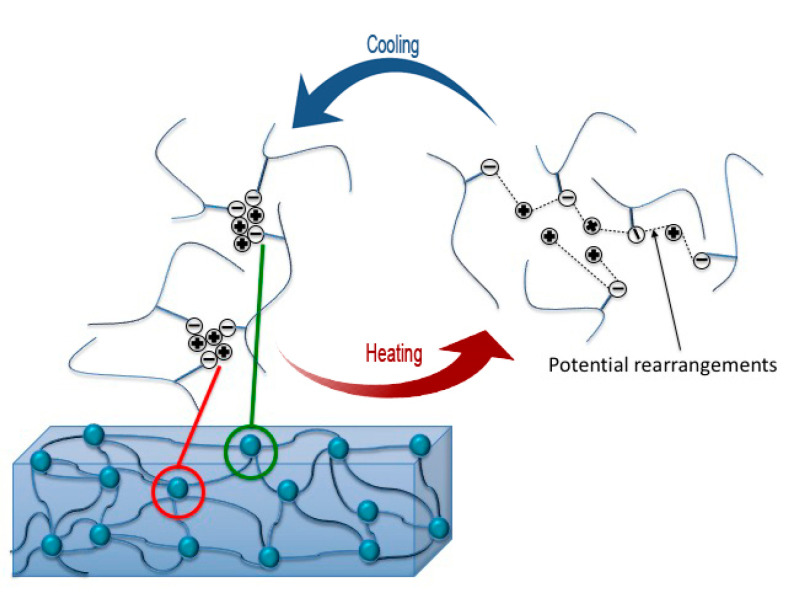
Ionomer structure with a focus on the ionic aggregate clusters that are tightly packed on the left-hand side. Upon heating, they have a larger margin for movement and can form new thermally reversible crosslinks.

**Figure 2 polymers-14-03575-f002:**
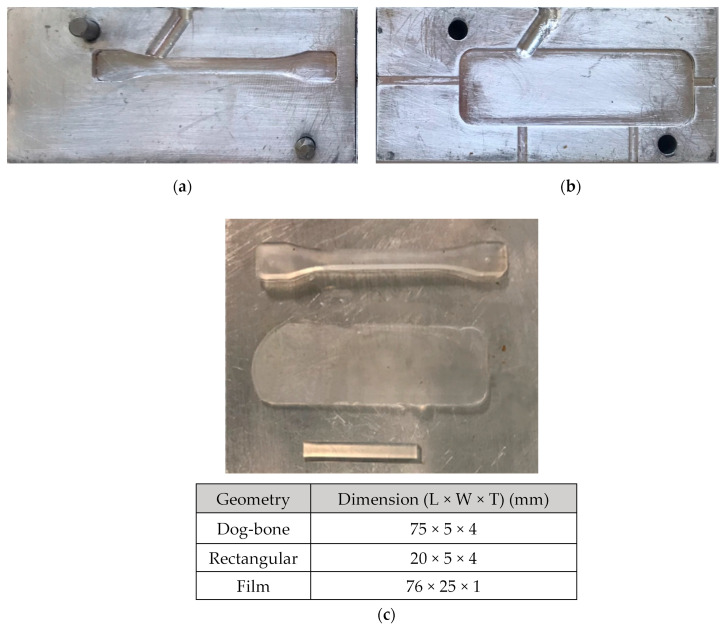
(**a**) The dog-bone mold and (**b**) the film mold and (**c**) the dog-bone specimen, the film, and the rectangular specimen, from top to bottom with their respective dimensions, used for characteristic tests and projectile tests of PEMAA co-polymers.

**Figure 3 polymers-14-03575-f003:**
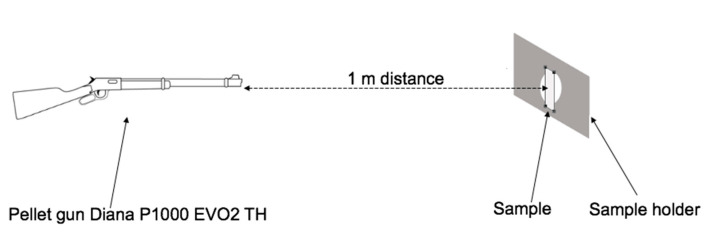
Schematic of the ballistic test with a pellet gun, a sample placed in a holder.

**Figure 4 polymers-14-03575-f004:**
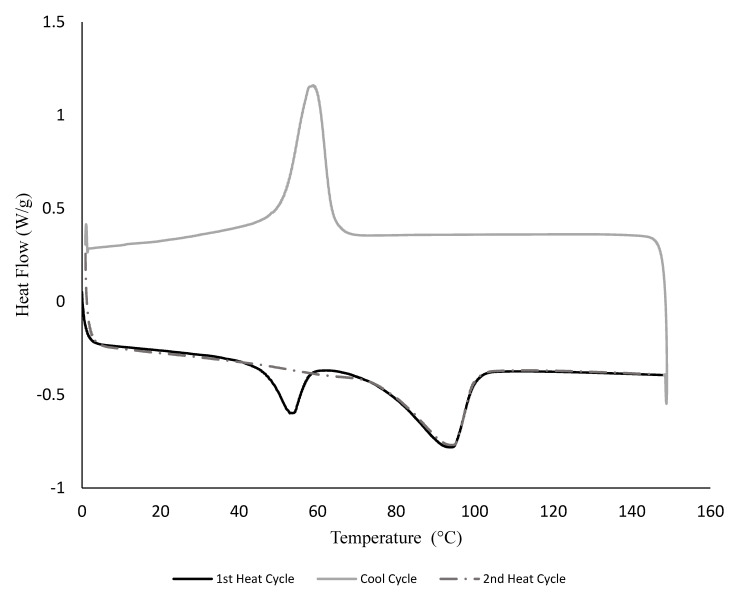
A DSC thermogram for a PEMAA-53%Li-M122 during a heat-cool-heat cycle at a rate of 5 °C/min.

**Figure 5 polymers-14-03575-f005:**
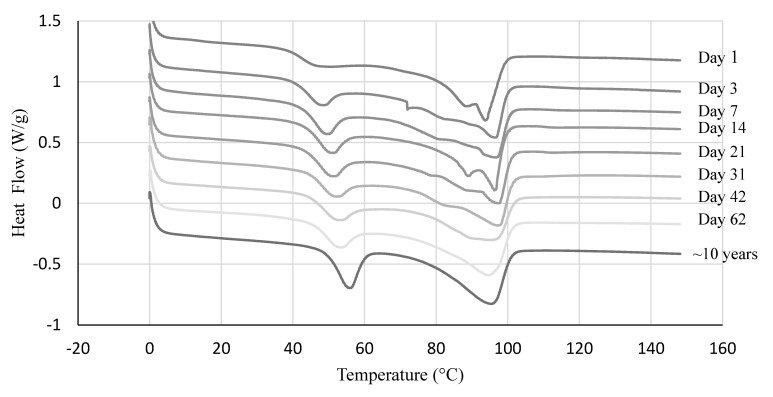
DSC thermograms of PEMAA-26%Li-122M at different time intervals from the date of injection molding at a time scale from 1 day to about 10 years.

**Figure 6 polymers-14-03575-f006:**
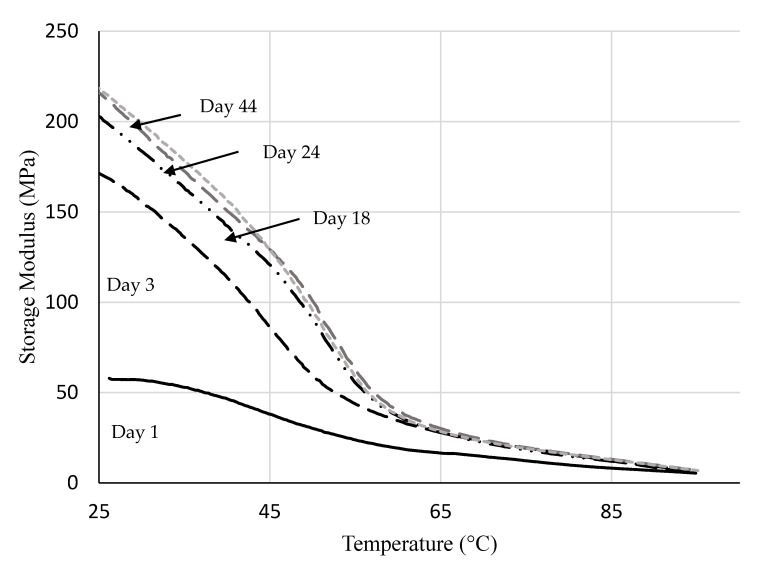
Mechanical aging test of PEMAA-26%Li-122M that shows the effect of aging on the storage modulus of PEMAA using DMA. Days refer to the time between the DMA run and when the sample was injected molded.

**Figure 7 polymers-14-03575-f007:**
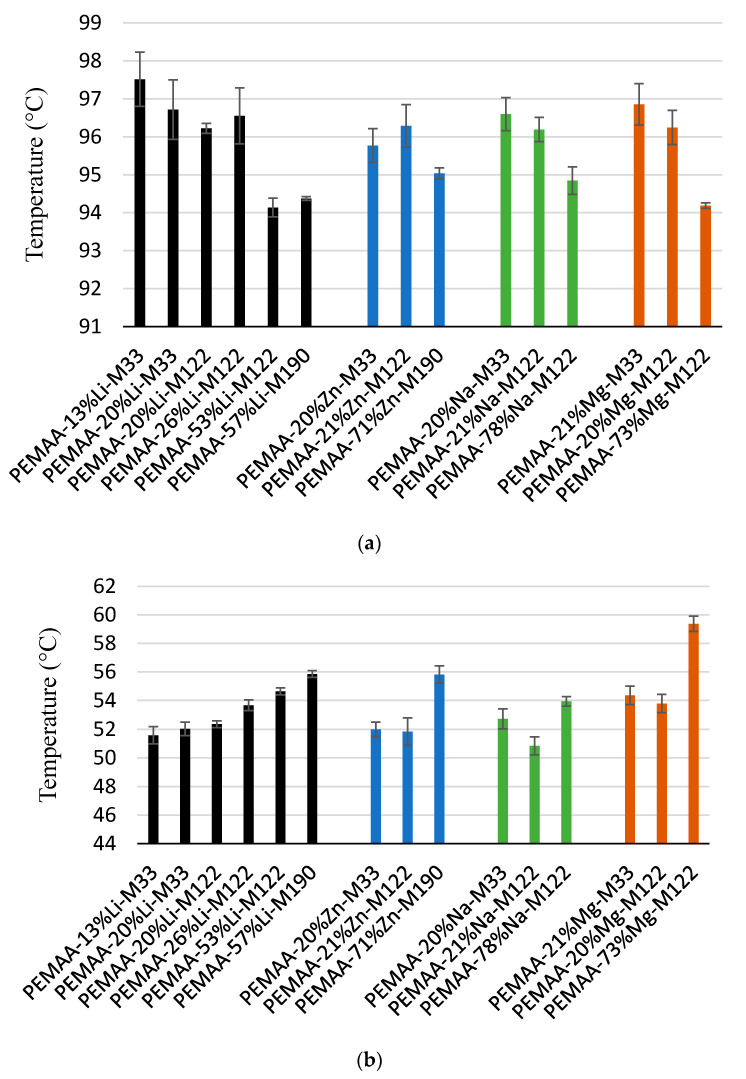
(**a**) Melting temperatures and (**b**) secondary crystallization temperatures of the PEMAA copolymers obtained from DSC tests.

**Figure 8 polymers-14-03575-f008:**
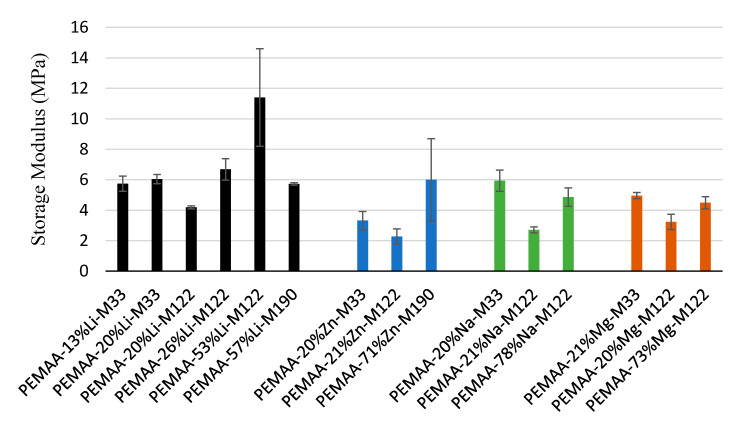
Storage modulus at 94.5 °C for all PEMAA copolymers under investigation from DMA tests.

**Figure 9 polymers-14-03575-f009:**
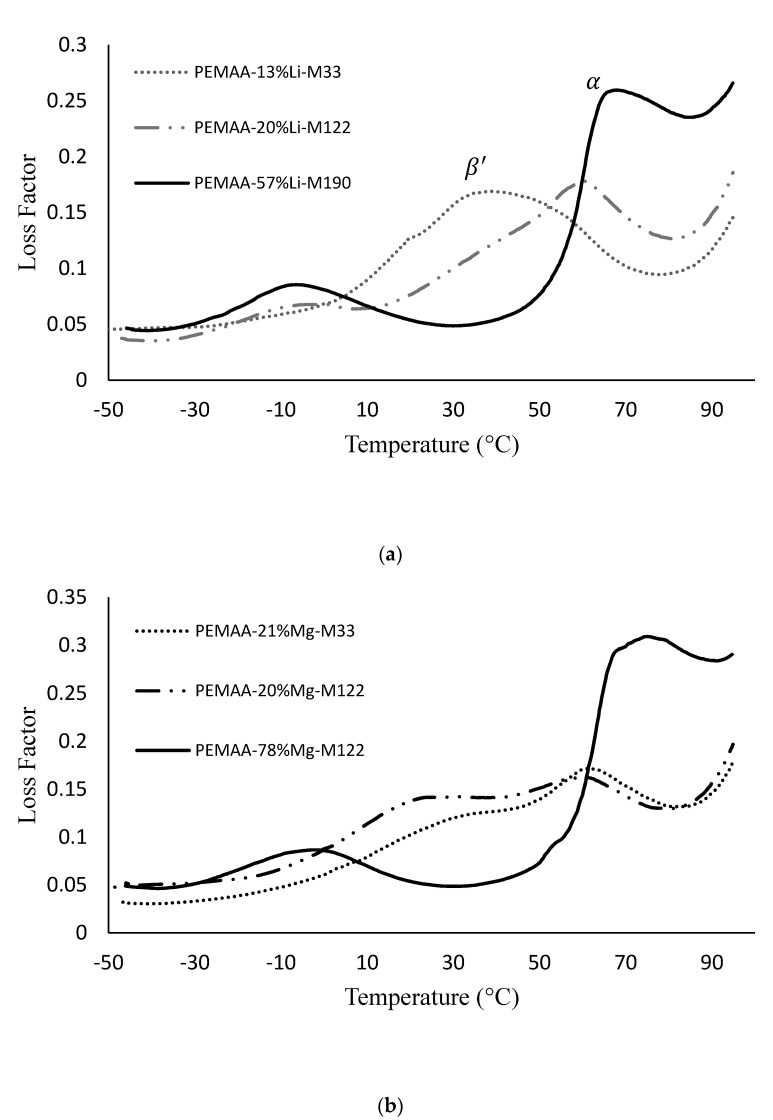
The loss factor obtained from DMA run for PEMAA neutralized at various percentages by (**a**) Li, (**b**) Mg, (**c**) Zn, and (**d**) Na.

**Figure 10 polymers-14-03575-f010:**
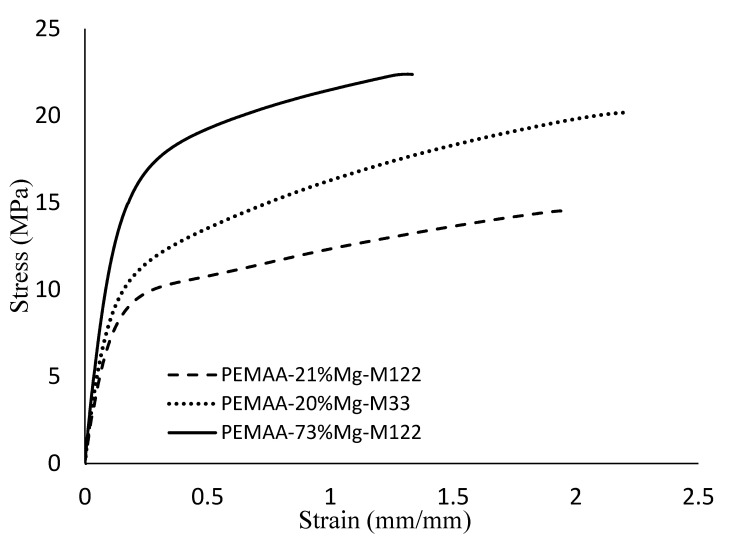
Representative graph of tensile stress vs. tensile strain for PEMAA-73%Mg-M122 and PEMAA-21%Mg-M122 and PEMAA-20%Mg-M33.

**Figure 11 polymers-14-03575-f011:**
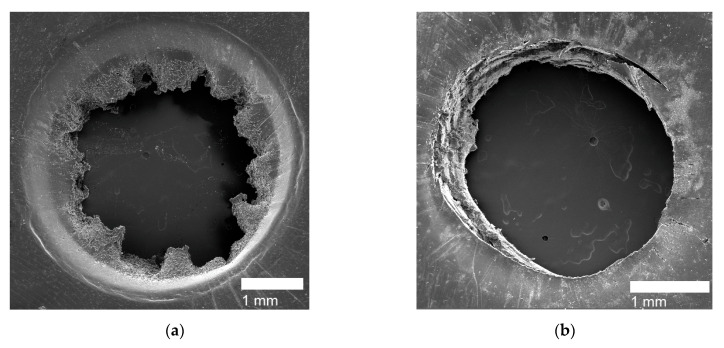
SEM image (**a**) shows the high impact response of the PE sample and SEM image (**b**) shows the circular holes of PEMAA-21%Zn-M122 at the impact site.

**Figure 12 polymers-14-03575-f012:**
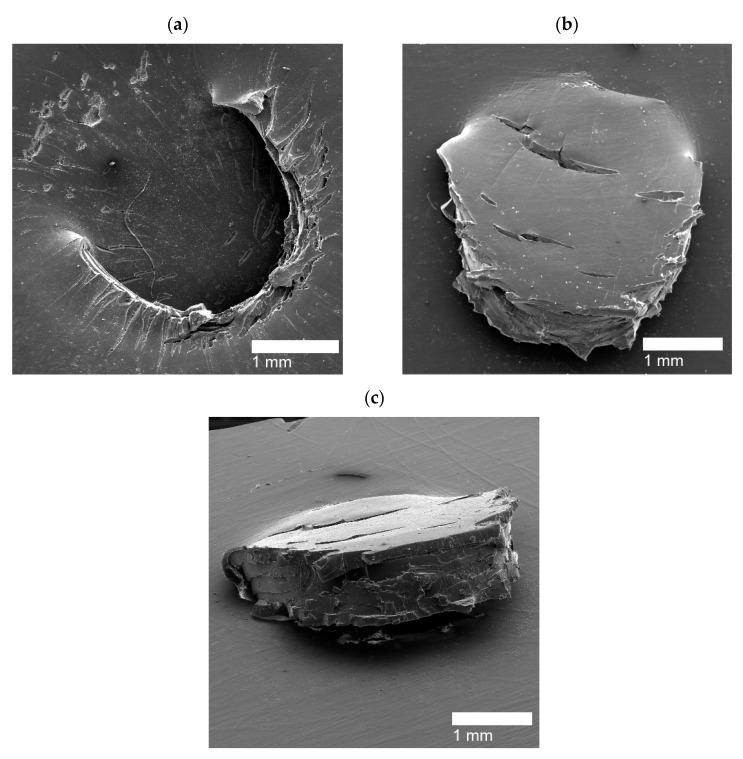
SEM image (**a**) shows the entry side, SEM image (**b**) shows the exit side, and SEM image (**c**) shows the diagonal view of the exit side of the door-flap puncture site for PEMAA-20%Li-M122.

**Figure 13 polymers-14-03575-f013:**
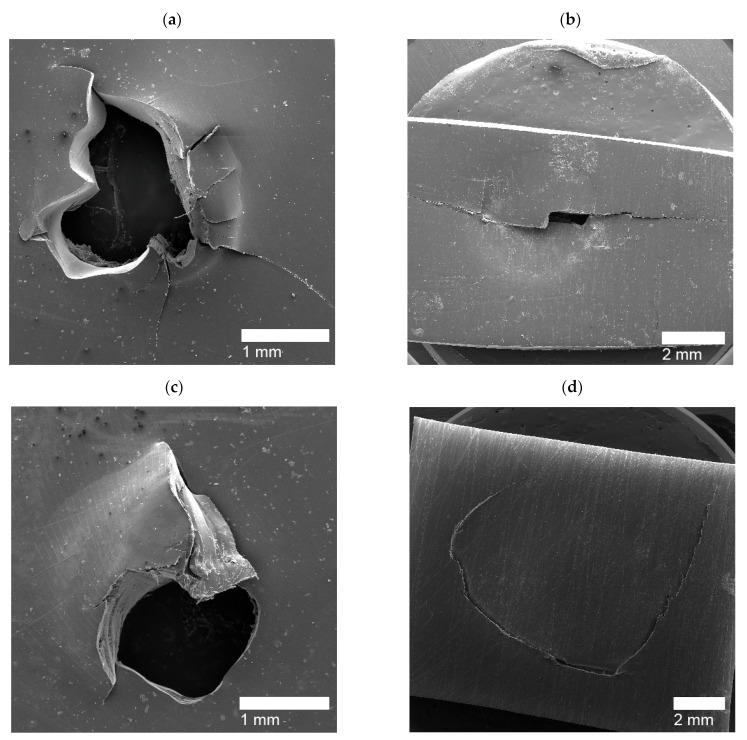
(**a**) Brittle hole from a PEMAA-71%Zn-M190 sample (**b**) Line fracture from a PEMAA-71%Zn-M190 sample (**c**) Brittle hole from of PEMAA-73%Mg-M122 sample (**d**) Line fracture from a PEMAA-73%Mg-M122 sample.

**Figure 14 polymers-14-03575-f014:**
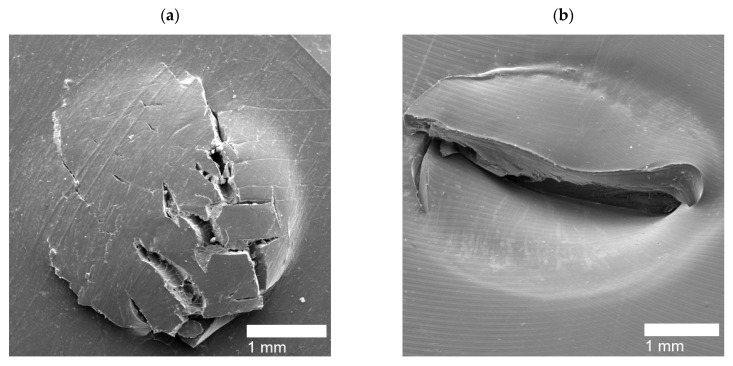
(**a**) Sealed exit site from a PEMAA-53%Li-M122 sample (**b**) Sealed entry site from a PEMAA-58%Li-M190 sample, and (**c**) Line fracture from a PEMAA-58%Li-M190 sample.

**Figure 15 polymers-14-03575-f015:**
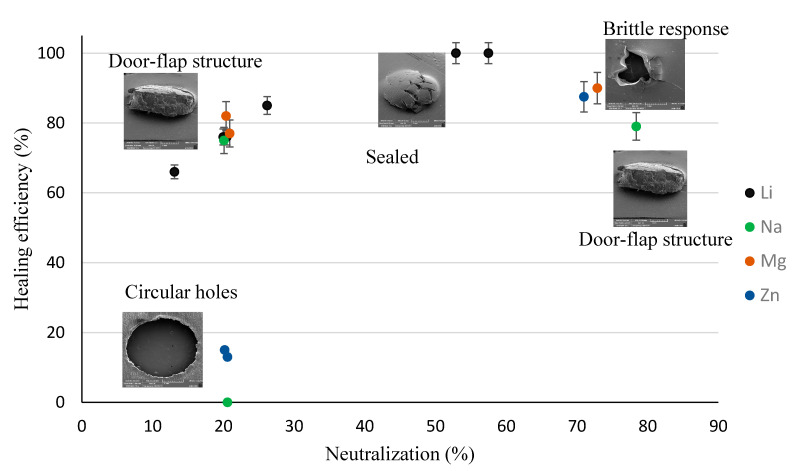
Healing efficiency obtained using Equation (1) as a function of neutralization percentage and ion type.

**Figure 16 polymers-14-03575-f016:**
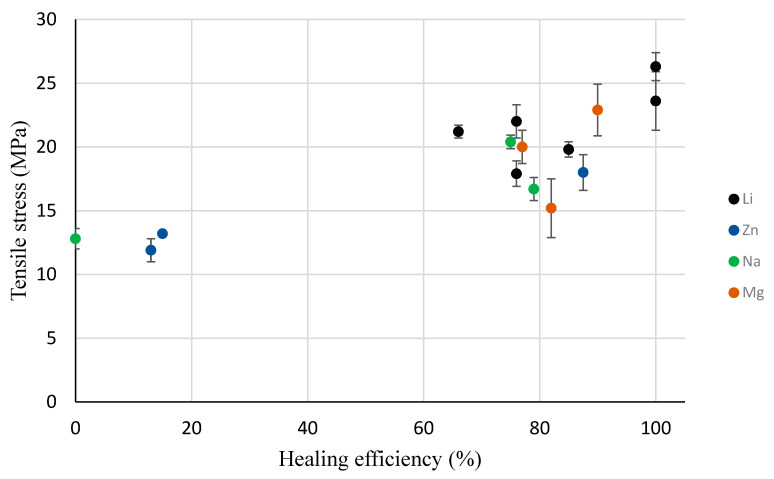
Tensile stress at maximum load for all PEMAA polymers at 25 °C in relation to the healing efficiency.

**Figure 17 polymers-14-03575-f017:**
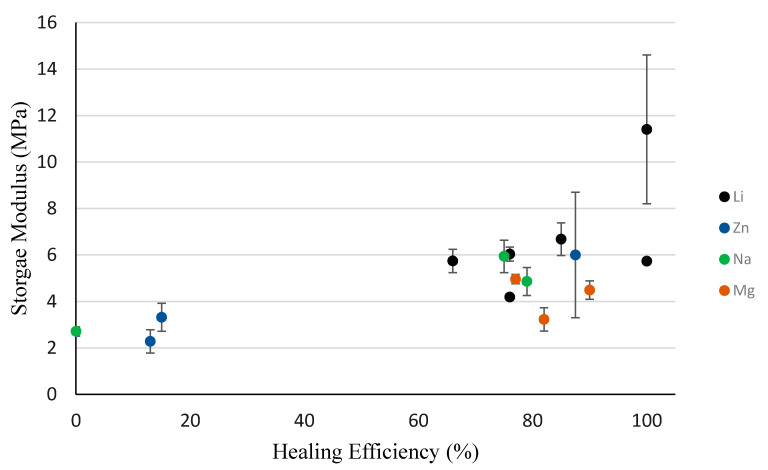
Storage Modulus at 94.5 °C for all PEMAA polymers in relation to the healing efficiency.

**Table 1 polymers-14-03575-t001:** List of polymers with the acid type of methacrylic acid (MAA), base melt index, ion type, and percentage neutralized.

Sample Name	Base Melt Index (g/10 min)	Ion Type	Neutralized %
PEMAA-13%Li-M33	33	Li	13.1%
PEMAA-20%Li-M33	33	Li	20.0%
PEMAA-20%Li-M122	122	Li	20.5%
PEMAA-26%Li-M122	122	Li	26.2%
PEMAA-53%Li-M122	122	Li	52.9%
PEMAA-57%Li-M190	190	Li	57.5%
PEMAA-20%Zn-M33	33	Zn	20.2%
PEMAA-21%Zn-M122	122	Zn	20.6%
PEMAA-71%Zn-M190	190	Zn	71.0%
PEMAA-20%Na-M33	33	Na	20.1%
PEMAA-21%Na-M122	122	Na	20.6%
PEMAA-78%Na-M122	122	Na	78.4%
PEMAA-21%Mg-M33	33	Mg	20.9%
PEMAA-20%Mg-M122	122	Mg	20.4%
PEMAA-73%Mg-M122	122	Mg	72.9%

**Table 2 polymers-14-03575-t002:** Base Ionomer Properties of melt index and mol % of copolymers ethylene (E) and methacrylic acid (MAA).

PEMAA	
Melt Index (g/10 min)	33, 122 (All), 190 (Zn, Li)
Mol % of the copolymer	E(96.35%), MAA(3.65%) (Low MI) E(96.4%), MAA(3.60%) (High MI)

**Table 3 polymers-14-03575-t003:** Tensile testing results for the 15 PEMAA copolymers.

Sample Name	Maximum Load [N]	Tensile Stress at Maximum Load [MPa]	Modulus [MPa]	Extension (%)
PEMAA-13%Li-M33	441 ± 18	21.2 ± 0.5	98 ± 2	2.60 ± 0.25
PEMAA-20%Li-M33	459 ± 24.9	22.0 ± 1.3	124 ± 11	1.67 ± 0.27
PEMAA-20%Li-M122	367 ± 14	17.9 ± 1.0	117 ± 3	1.88 ± 0.15
PEMAA-26%Li-M122	413 ± 16	19.8 ± 0.6	111 ± 4	1.45 ± 0.13
PEMAA-53%Li-M122	517 ± 27	26.3 ± 1.1	144 ± 9	1.15 ± 0.17
PEMAA-57%Li-M190	484 ± 58	23.6 ± 2.3	143 ± 7	1.43 ± 0.08
PEMAA-20%Zn-M33	279 ± 9	13.2 ± 0.2	67.5 ± 5.9	1.82 ± 0.07
PEMAA-21%Zn-M122	235 ± 16	11.9 ± 0.9	69.1 ± 9.8	2.12 ± 0.37
PEMAA-71%Zn-M190	368 ± 23	18.0 ± 1.4	200 ± 2	1.60 ± 0.33
PEMAA-20%Na-M33	418 ± 13	20.4 ± 0.53	85.9 ± 8.4	2.02 ± 0.14
PEMAA-21%Na-M122	250 ± 18	12.8 ± 0.8	102 ± 13	1.63 ± 0.21
PEMAA-78%Na-M122	340 ± 27	16.7 ± 0.9	72.6 ± 6.8	1.51 ± 0.10
PEMAA-21%Mg-M33	425 ± 21	20.0 ± 1.3	93.7 ± 11.4	2.45 ± 0.27
PEMAA-20%Mg-M122	309 ± 33	15.2 ± 2.3	89.0 ± 10.8	2.10 ± 0.33
PEMAA-73%Mg-M122	484 ± 50	22.9 ± 2.03	135 ± 3	1.44 ± 0.21

**Table 4 polymers-14-03575-t004:** The responses of PEMAA samples to the high-impact puncture.

Puncture Type	Sample Name
Circular Holes	PE, PEMAA-21%Na-M122, PEMAA-21%Zn-M122, PEMAA-20%Zn-M33
Door-flap	PEMAA-13%Li-M33, PEMAA-20%Li-M33, PEMAA-20%Li-M122, PEMAA-26%Li-M122, PEMAA-20%Na-M33, PEMAA-78%Na-M122, PEMAA-21%Mg-M33, PEMAA-20%Mg-M122
Brittle Holes	PEMAA-71%Zn-M190, PEMAA-73%Mg-M122
Line Fracture	PEMAA-71%Zn-M190, PEMAA-57%Li-M190, PEMAA-73%Mg-M122
Sealed site	PEMAA-53%Li-M122, PEMAA-57%Li-M190

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
