# Peer review of "Self-Healability of Poly(Ethylene-co-Methacrylic Acid): Effect of Ionic Content and Neutralization"

_polymers, 2022, doi:10.3390/polym14173575_

Round 1

Reviewer 1 Report

1.     The first line of the introduction section “Self-healing polymers can recover their initial performance after damage partially or fully without the need for significant human intervention and repair” should be corrected to “Self-healing polymers can recover their initial performance after being damaged partially or fully without the need for significant human intervention and repair”

2. Similar grammatical mistakes should be checked and corrected. 

3. The ion percentage used to neutralize the PEMAA polymers is confusing. How did the authors chose the compositions? 

4. I did not understand why the authors took SEM images to show mm scale images.

Author Response

Reviewer 1

  1. The first line of the introduction section “Self-healing polymers can recover their initial performance after damage partially or fully without the need for significant human intervention and repair” should be corrected to “Self-healing polymers can recover their initial performance after being damaged partially or fully without the need for significant human intervention and repair”

- The first sentence was revised as follows:

“Self-healing polymers can partially or fully recover their initial performance after being damaged without the need for significant human intervention and repair”

  1. Similar grammatical mistakes should be checked and corrected. 

The following sentence was revised in the third paragraph of the Introduction:

“Capsular self-healing polymer-based systems introduce healing agents in microcapsules that crack after getting damaged”

- The first sentence of the second last paragraph of the Conclusion was revised as follows:

“At high neutralization percentages of 70% and above, DMA and tensile testing showed that the PEMAA polymers become increasingly stiff and brittle.”

- The last paragraph of the Conclusion was revised as follows:

“Mid-range neutralized PEMAA samples were able to self-heal and seal the puncture site after being shot with an air rifle. This behavior was attributed to their viscoelastic properties to allow enough elasticity to allow an instantaneous snapback mechanism where the molten edges sealed the puncture site.”

- The tense was changed in the following sentence on page 13:

“The storage modulus, which is the ratio of elastic stress to strain during dynamic mechanical analysis, measures the amount of energy that the specimen can store elastically”

  1. The ion percentage used to neutralize the PEMAA polymers is confusing. How did the authors chose the compositions? 

- As we mentioned in 2.1. Materials, 15 PEMAA copolymers were produced and supplied by the former DuPont Chemical Company (now part of Dow) as shown in Table 1. Percentages represent the fraction of acid groups that have been neutralized by metal salts of Li, Na, Mg, or Zn. These were chosen to fulfill three criteria (1)  Be able to compare different MI resins (33 and 122) at the same neutralization percentage and same cation(2) Be able to compare a high (~75%) and low neutralization level (~20%) for the same base resin and cation.  (3) Be able to compare different cations Li, Na, Mg, or Zn. at the same neutralization level and base resin.   Unfortunately, the high for the Li resin, for reasons that we aren’t clear was only ~55% instead of ~75%.  

  1. I did not understand why the authors took SEM images to show mm scale images.

- We took the images with an SEM to obtain the highest image quality and resolution.  We could have alternatively taken the pictures with an optical microscope.

Reviewer 2 Report

I have reviewed the manuscript entitled " Self-healability of Poly (Ethylene-co-Methacrylic Acid): Effect of Ionic Content and Neutralization " carefully. The authors studied the thermal-mechanical properties and self-healing response of 15 PEMAA copolymers with a variety of ionic content (Li, Na, Zn, Mg) ) and neutralization ( (13 to 78%). The paper showed the choice of ion and the amount of neutralization affected self-healing responses in PEMAA copolymers. I think the paper was well written with a clear purpose, the experiments were well designed, and the results were clearly presented. The paper can be published.

Author Response

We would like to thank you for reviewing the manuscript and highlighting the strength of the work.